# Incremental Predictive Coding: A Parallel and Fully Automatic Learning Algorithm

## Abstract

Neuroscience-inspired models, such as predictive coding, have the potential to play an important role in the future of machine intelligence. However, they are not yet used in industrial applications due to some limitations, one of them being the lack of efficiency. In this work, we address this by proposing incremental predictive coding (iPC), a variation of the original framework derived from the incremental expectation maximization algorithm, where every operation can be performed in parallel without external control. We show both theoretically and empirically that iPC is more efficient than the original algorithm by Rao and Ballard 1999, while maintaining performance comparable to backpropagation in image classification tasks. This work impacts several areas, has general applications in computational neuroscience and machine learning, and specific applications in scenarios where automatization and parallelization are important, such as distributed computing and implementations of deep learning models on analog and neuromorphic chips.

## 1 Introduction

In recent years, deep learning has reached and surpassed human-level performance in a multitude of tasks, such as game playing (Silver et al., 2017; 2016), image recognition (Krizhevsky et al., 2012; He et al., 2016), natural language processing (Chen et al., 2020), and image generation (Ramesh et al., 2022). These successes are achieved entirely using deep artificial neural networks trained via *backpropagation* (*BP*), which is a learning algorithm that is often criticized for its biological implausibilities (Grossberg, 1987; Crick, 1989; Abdelghani et al., 2008; Lillicrap et al., 2016; Roelfsema & Holtmaat, 2018; Whittington & Bogacz, 2019), such as lacking local plasticity and autonomy. In fact, backpropagation requires a global control signal required to trigger computations, since gradients must be sequentially computed backwards through the computation graph. These properties are not only important for biological plausibility: *parallelization*, *locality*, and *automation* are key to build efficient models that can be trained end-to-end on non Von-Neumann machines, such as analog chips (Kendall et al., 2020). A learning algorithm with most of the above properties is predictive coding (PC).

PC is an influential theory of information processing in the brain (Mumford, 1992; Friston, 2005), where learning happens by minimizing the prediction error of every neuron. PC can be shown to approximate backpropagation in layered networks (Whittington & Bogacz, 2017), as well as on any other model (Millidge et al., 2020), and can exactly replicate its weight update if some external control is added (Salvatori et al., 2022b). Also the differences with BP are interesting, as PC allows for a much more flexible training and testing (Salvatori et al., 2022a), has a rich mathematical formulation (Friston, 2005; Millidge et al., 2022), and is an energy-based model (Bogacz, 2017). This makes PC unique, as it is the only model that jointly allows training on neuromorphic chips, is an implementation of influential models of cortical functioning in the brain, and can match the performance of backpropagation in different tasks. Its main drawback, however, is the efficiency, as it is slower than BP. In this work, we address this problem by proposing a variation of PC that is much more efficient than the original one.

Simply put, PC is based on the assumption that brains implement an internal generative model of the world, needed to predict incoming stimuli (or data) (Friston et al., 2006; Friston, 2010; Friston et al., 2016). When presented with a stimulus that differs from the prediction, learning happens by

updating internal neural activities and synapses to minimize the *prediction error*. In computational models, this is done via multiple expectation-maximization (EM) (Dempster et al., 1977) steps on the variational free energy, in this case a function of the total error of the generative model. During the E-step, internal neural activities are updated in parallel until convergence; during the M-step, a weight update to further minimize the same energy function is performed. This approach results in two limitations: first, the E-step is slow, as it can require dozens of iterations before convergence; second, an external control signal is needed to switch from the E to M step. In this paper, we show how to address both of these problems by considering a variation of the EM algorithm, called *incremental expectation-maximization* (iEM), which performs both E and M steps in parallel (Neal & Hinton, 1998). This algorithm is provably faster, does not require a control signal to switch between the two steps, and has solid convergence guarantees (Neal & Hinton, 1998; Karimi et al., 2019). What results is a training algorithm that we call *incremental predictive coding* (iPC) that is a simple variation of PC that addresses the main drawback of PC (namely, efficiency), with no drawbacks from the learning perspective, as it has been formally proven to have equivalent convergence properties to standard PC. Furthermore, we provide initial evidence that iPC is also potentially more efficient than BP in the specific case of full-batch training. In fact, we theoretically show that, on an ideal parallel machine, to complete one update of all weights on a network with $L$ layers, the time complexity of iPC is $\mathcal{O}(1)$, while that of BP is $\mathcal{O}(L)$. However, additional engineering efforts are needed to reach this goal, which are beyond the focus of this work: our experiments are performed using PyTorch (Paszke et al., 2017), which is not designed to parallelize computations across layers on GPUs. We partially address this limitation by performing some experiments on CPUs, which empirically confirm our claims about efficiency, as shown in Fig. 3. Our contributions are briefly as follows:

1. We first develop the update rule of iPC from the variational free energy of a hierarchical generative model using the incremental EM approach. We then discuss the implications of this change in terms of autonomy and convergence guarantees: it has in fact been proven that iEM converges to a minimum of the loss function (Neal & Hinton, 1998; Karimi et al., 2019), and hence this result naturally extends to iPC. We conclude by analyzing similarities and differences between iPC, standard PC, and BP.

2. We empirically compare the efficiency of PC and iPC on generation tasks, by replicating some experiments performed in (Salvatori et al., 2021), and classification tasks, by replicating experiments similar to those presented in (Whittington & Bogacz, 2017). In both cases, iPC is by far more efficient than the original counterpart. Furthermore, we present initial evidence that iPC can decrease the training loss faster than BP, assuming that a proper parallelization is done.

3. We then test our model on a large number of image classification benchmarks, showing that that iPC performs better than PC, on average, and similarly to BP. Then, we show that iPC requires less parameters than BP to perform well on convolutional neural networks (CNNs). Finally, we show that iPC follows the trends of energy-based models on training robust classifiers (Grathwohl et al., 2019), and yields better calibrated outputs than BP on the best performing models.

## 2 PRELIMINARIES

In this section, we introduce the original formulation of predictive coding (PC) as a generative model proposed by Rao and Ballard 1999. Let us consider a generative model $g : \mathbb{R}^d \times \mathbb{R}^D \longrightarrow \mathbb{R}^o$, where $x \in \mathbb{R}^d$ is a vector of latent variables called *causes*, $y \in \mathbb{R}^o$ is the generated vector, and $\theta \in \mathbb{R}^D$ is a set of parameters. We are interested in the following inverse problem: given a vector $y$ and a generative model $g$, we need the parameters $\theta$ that maximize the marginal likelihood

$$p(y, \theta) = \int_x p(y \mid x, \theta) p(x, \theta) dx. \tag{1}$$

Here, the first term inside the integral is the likelihood of the data given the causes, and the second is a prior distribution over the causes. Solving the above problem is intractably expensive. Hence, we need an algorithm that is divided in two phases: *inference*, where we infer the best causes $x$ given both $\theta$ and $y$, and *learning*, where we update the parameters $\theta$ based on the newly computed causes. This algorithm is *expectation-maximization* (EM) (Dempster et al., 1977). The first step, which we call inference or E-step, computes $p(x \mid y, \theta)$, that is the posterior distribution of the causes given

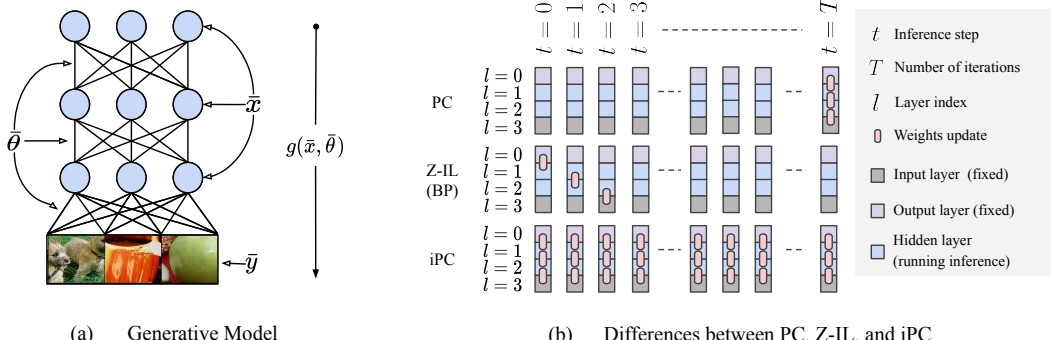

| | (a) Generative Model | | (b) Differences between PC, Z-IL, and iPC |

Figure 1: (a) An example of an hierarchical Gaussian generative model with three layers. (b) Comparison of the temporal training dynamics of PC, Z-IL, and iPC, where Z-IL is a variation of predictive coding that is equivalent to BP, originally introduced in Song et al. (2020). We assume that we train the networks on a dataset for supervised learning for a period of time $T$. Here, $t$ is the time axis during inference, which always starts at $t = 0$. The squares represent nodes in one layer, and pink rounded rectangles indicate when the connection weights are modified: PC (1st row) first conducts inference on the hidden layers, according to Eq. equation 6, until convergence, and then it updates the weights via Eq. 7. Z-IL (2nd row) only updates the weights at specific inference moments depending on which layer the weights belong to. To conclude, iPC updates the weights at every time step $t$, while performing inference in parallel.

a generated vector $y$. Computing the posterior is, however, intractable (Friston, 2003). To this end, we approximate the intractable posterior with a tractable probability distribution $q(x, \theta)$. To make the approximation as good as possible, we want to minimize the KL-divergence between the two probability distributions. Summarizing, to solve our learning problem, we need to (i) minimize a KL-divergence, and (ii) maximize a likelihood. We do it by defining the following energy function, also known as *variational free-energy*:

$$F(x, y, \theta) = KL(q(x, \theta) \parallel p(x \mid y, \theta)) - ln(p(y, \theta)), \tag{2}$$

where we have used the log-likelihood. This function is minimized by multiple iterations of the EM algorithm as follows:

$$\begin{cases} \text{Inference (E-step): } x^* = argmax_x F(x, y, \theta), \\ \text{Learning (M-step): } \theta^* = argmax_\theta F(x, y, \theta). \end{cases} \tag{3}$$

## 2.1 PREDICTIVE CODING

So far, we have only presented the general problem. To actually derive proper equations for learning causes and update the parameters, and use them to train neural architectures, we need to specify the generative function $g(x, \theta)$. Following the general literature, (Rao & Ballard, 1999; Friston, 2005), we define the generative model as a hierarchical Gaussian generative model, where the causes $x$ and parameters $\theta$ are defined by a concatenation of the causes and weight matrices of all the layers, i.e., $x = (x^{(0)}, \ldots, x^{(L)})$, and $\theta = (\theta^{(0)}, \ldots, \theta^{(L-1)})$. Hence, we have a multilayer generative model, where layer $0$ is the one corresponding to the generated image $y$, and layer $L$ the highest in the hierarchy. The marginal probability of the causes is as follows:

$$p(x^{(0)}, \ldots, x^{(L)}) = p(x^{(L)}) \prod_{l}^{L-1} p(x^{(l-1)} \mid x^{(l)}) = \prod_{l}^{L} \mathcal{N}(\mu^{(l)}, \Sigma^{(l)}), \tag{4}$$

where $\mu^{(l)}$ is the *prediction* of layer $l$ according to the layer above, given by $\mu^{(l)} = \theta^{(l)} \cdot f(x^{(l+1)})$, with $f$ being a non-linear function and $\mu^{(L)} = x^{(L)}$. For simplicity, from now on, we consider Gaussians with identity variance, i.e., $\Sigma^{(l)} = \mathbb{1}$ for every layer $l$. With the above assumptions, the

---

**Algorithm 1** Learning a dataset $\mathcal{D} = \{y_i\}$ with iPC.

---

1: **Require:** For every $i$, $x_i^{(0)}$ is fixed to $y_i$,
2: **for** $t = 0$ to $T$ **do**
3:     For every $i$ and $l$, update $x_i^{(l)}$ to minimize $F$ via Eq.(6)
4:     For every $l$, update each $\theta^{(l)}$ to minimize $F$ via Eq.(7)
5: **end for**

---

free-energy becomes

$$F = \sum_l \|x^{(l)} - \mu^{(l)}\|^2. \tag{5}$$

For a detailed formulation on how this energy function is derived from the variational free-energy of Eq. 2, we refer to (Friston, 2005; Bogacz, 2017; Buckley et al., 2017), or to the supplementary material. Note that this energy function is equivalent to the one proposed in the original formulation of predictive coding (Rao & Ballard, 1999). A key aspect of this model is that both inference and learning are achieved by optimizing the same energy function, which aims to minimize the prediction error of the network. The prediction error of every layer is given by the difference between its real value $x^{(l)}$ and its prediction $\mu^{(l)}$. We denote the prediction error $\varepsilon^{(l)} = x^{(l)} - \mu^{(l)}$. Thus, the problem of learning the parameters that maximize the marginal likelihood given a data point $y$ reduces to an alternation of inference and weight update. During both phases, the values of the last layer are fixed to the data point, i.e., $x^{(0)} = y$ for every $t \leq T$.

**Inference:** During this phase, that corresponds to the E-step, the weight parameters $\theta^{(l)}$ are fixed, while the values $x^{(l)}$ are continuously updated via gradient descent.

$$\Delta x^{(l)} = -\gamma \frac{\partial F}{\partial x^{(l)}} = \gamma \cdot (-\varepsilon^{(l)} + f'(x^{(l)}) * \theta^{(l-1)\,\mathsf{T}} \cdot \varepsilon^{(l-1)}), \tag{6}$$

where $*$ denotes element-wise multiplication, and $l > 0$. This process either runs until convergence, or for a fixed number of iterations $T$.

**Learning:** During this phase, which corresponds to the M-step, the values $x$ are fixed, and the weights are updated once via gradient descent according to the following equation:

$$\Delta\theta^{(l)} = -\alpha \frac{\partial F}{\partial \theta^{(l)}} = \alpha \cdot x^{(l+1)} \varepsilon^{(l)}. \tag{7}$$

Note that the above algorithm is not limited to generative tasks, but can also be used to solve supervised learning problems (Whittington & Bogacz, 2017). Assume that we are provided with a data point $y_{in}$ with label $y_{out}$. In this case, we treat the label as the vector $y$ we need to generate, and the data point as the prior on $x^{(L)}$. The inference and learning phases are identical, with the only difference that now we have two vectors fixed during the whole duration of the process: $x^{(0)} = y_{out}$, and $x^{(L)} = y_{in}$. While this algorithm is able to obtain good results on small image image classification tasks, it is much slower than BP due to the large number of inference steps $T$ needed to let the causes $x$ converge.

## 3 INCREMENTAL PREDICTIVE CODING

One of the main drawbacks of energy based models such as PC and equilibrium propagation (Scellier & Bengio, 2017), is their efficiency. In fact, these algorithms are much slower than BP due to the inference phase, which requires multiple iterations to converge. The goal of this paper is to address this problem for predictive coding, by developing a variation based from the *incremental* EM (iEM) algorithm (Neal & Hinton, 1998), which was developed to address the lack of efficiency of the original EM. This algorithm excels when dealing with multiple data points at the same time (Neal & Hinton, 1998), a scenario that is almost always present in standard machine learning.

Let $\mathcal{D} = \{y_i\}_{i<N}$ be a dataset of cardinality $N$, and $g(x, \theta)$ be a generative model. Our goal is now to minimize the global marginal likelihood, defined on the whole dataset, i.e.,

$$p(\mathcal{D}, \theta) = \sum_i p(y_i, \theta). \tag{8}$$

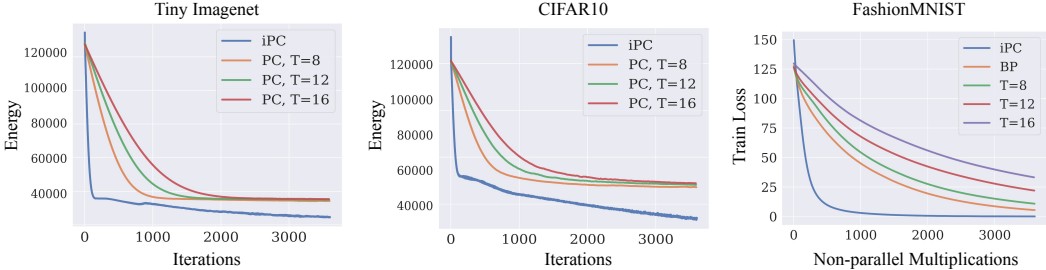

Figure 2: Left and centre: Decrease of the energy of generative models as a function of the number of iterations performed from the beginning of the training process. Right: Training loss of different classifiers trained using iPC, BP, and multiple parameterizations of PC as a function of the number of non-parallel matrix multiplications performed from the beginning of the training process.

The same reasoning also applies to the global variational free energy, which will be the sum of the free energies of every single data point. In this case, the iEM algorithm performs the E-step and M-step in parallel, with no external control needed to switch between the two phases. In detail, both the values $x$ and the parameters $\theta$ are updated simultaneously at every time step $t$, until convergence (or for a fixed number of iterations $T$), according to the same update rule defined in Eqs. 6 and 7, on all the points of the dataset. No explicit forward and backward passes are necessary as each layer is updated in parallel. To our knowledge, this is the first learning algorithm for deep neural networks where every single operation is performed in parallel. Note that this increased speed does not harm the final performance, as it has been formally proven that minimizing a free-energy function such as ours (i.e., equivalent to the sum of independent free-energy functions) using iEM, also finds a minimum of the global marginal likelihood of Eq.8 (Neal & Hinton, 1998; Karimi et al., 2019). We actually provide empirical evidence that the model converges to better minima using iPC rather than the original formulation of PC in Fig. 2 and Table 1. The pseudocode of iPC is given in Alg. 1.

**Connections to BP:** PC in general shares multiple similarities with BP in supervised learning tasks: when the output error is small, the parameter update of PC is an approximation of that of BP (Millidge et al., 2020); when controlling which parameters have to be updated at which time step, the two updates can even be made equivalent (Salvatori et al., 2022b). To make PC perform exactly the same weight update of BP, every weight matrix $\theta^l$ must be updated only at $t = l$, which corresponds to its position in the hierarchy (Song et al., 2020). That is, as soon as the output error reaches a specific layer. This is different from the standard formulation of PC, which updates the parameters only when the energy representing the total error has converged. Unlike PC, iPC updates the parameters at every time step $t$. Intuitively, it can hence be seen as a "continuous shift" between Z-IL and PC, where Z-IL is a variation of PC that is equivalent to BP, originally introduced in Song et al. (2020).. A graphical representation of the differences of all three algorithms is given in Fig. 1 (right), with the pseudo-codes provided in the first section of the supplementary material.

**Autonomy:** Both PC and Z-IL lack full autonomy, as an external control signal is always needed to switch between inference and learning: PC waits for the inference to converge (or, for $T$ iterations), while Z-IL updates the weights of specific layers at specific inference moments $t = l$. BP is considered to be less autonomous than PC and Z-IL: a control signal is required to forward signals as well as backward errors, and additional places to store the backward errors are required. All of those drawbacks are removed in iPC, which is able to learn a dataset without the control signals required by the other algorithms: given a dataset $\mathcal{D}$, iPC runs inference and weight updates simultaneously until the energy $F$ is minimized. As soon as the energy minimization has converged, training ends.

## 3.1 EFFICIENCY

In this section, we analyze the efficiency of iPC with respect to both the original formulation of PC and BP. We only provide partial evidence of the increased efficiency against BP, as standard deep learning frameworks, such as Pytorch, do not allow to parallelize operations in different layers. While we leave the development of a framework able to perform every operation in parallel to future work, we provide evidence that the speed up against BP in full batch training is theoretically possible using iPC.

**Comparison with PC:** We now show how iPC is more efficient than the original formulation. To do that, we have trained multiple models with iPC and PC on different tasks and datasets. First, we have trained a generative model with 4 layers and 256 hidden neurons on a subset of 100 images of the Tiny ImageNet and CIFAR10 datasets, as done in (Salvatori et al., 2021). A plot with the energies as a function of the number of iterations is presented in Fig. 2 (left and centre). In both cases, the network trained with iPC converges much faster than the networks trained with PC with different values of $T$. Many more plots with different parameterizations are given in Fig. 7 in the supplementary material.

To show that the above results hold in different set-ups as well, we have trained a classifier with 4 layers on a subset of 250 images of the FashionMNIST dataset, following the framework proposed in (Whittington & Bogacz, 2017), and studied the training loss. As it is possible to train an equivalent model using BP, we have done it using the same set-up and learning rate, and included it in the plot. This, however, prevents us from using the number of iterations as an efficiency measure, as one iteration of BP is more complex than one iteration of PC, and are hence not comparable. As a metric, we have hence used the number of non-parallel matrix multiplications needed to perform a weight update. This is a fair metric, as matrix multiplications are by far the most expensive operation performed when training neural networks, and the ones with largest impact on the training speed. One iteration of PC and iPC have the same speed, and consist of 2 non-parallel matrix multiplications. One epoch of BP, consists of $2L$ non-parallel matrix multiplications. The results are given in Fig. 2 (right). In all cases, iPC converges much faster than all the other methods. In the supplementary material, we provide other plots obtained with different datasets, models, and parameterizations, as well as a study on how the test error decreases during training. Again, many more plots with different parameterizations are given in Fig. 8 in the supplementary material.

**Comparison with BP:** While the main goal of this work is simply to overcome the core limitation of original PC — the slow inference phase — there is one scenario where iPC is potentially more efficient than BP, which is full batch training. Particularly, we first prove this formally using the number of non-parallel matrix multiplications needed to perform a weight update as a metric. To complete one weight update, iPC requires two sets of non-parallel multiplications: the first uses the values and weight parameters of every layer to compute the prediction of the layer below; the second uses the error and transpose of the weights to propagate the error back to the layer above, needed to update the values. BP, on the other hand, requires $2L$ sets of non-parallel multiplications for a complete update of the parameters: $L$ for a forward pass, and $L$ for a backward one. These operations cannot be parallelized. More formally, we prove a theorem that holds when training on the whole dataset $\mathcal{D}$ in a full-batch regime. For details about the proof, and an extensive discussion about time complexity of BP, PC, and iPC, we refer to the supplementary material.

**Theorem 1** *Let $M$ and $M'$ be two equivalent networks with $L$ layers trained on the same dataset. Let $M$ be trained using BP, and $M'$ be trained using iPC. Then, the time complexity needed to perform one full update of the weights is $\mathcal{O}(1)$ for iPC and $\mathcal{O}(L)$ for BP.*

### 3.2 CPU Implementation

To further provide evidence of the efficiency of iPC with respect to BP, we have implmented the parallelization of iPC on a CPU, and compared it to BP, also implemented on CPU. We compute the time in milliseconds (ms) needed to perform one weight update of both on a randomly generated datapoint. In Fig. 3, we have plotted the ratio

*ms of iPC / ms of BP*

for architectures with different depths and widths. The results show that our naive implementation adds a computational overhead given by communication and synchronization across threads that makes iPC slower than BP on small

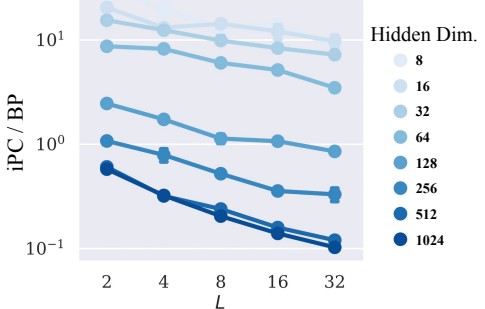

Figure 3: Ratio of the actual running time needed to perform a single weight update between BP and iPC on a CPU. Every dot represents a model, if the model lies below the horizontal line with label $10^0$, its weight update performed using iPC is faster than one performed using BP.

Table 1: Final accuracy of BP, PC, and iPC on different architectures trained with different datasets.

|  | BP/Z-IL | PC | iPC |
|---|---|---|---|
| MLP on MNIST | $98.26\% \pm 0.12\%$ | $\mathbf{98.55}\% \pm 0.14\%$ | $98.54\% \pm 0.86\%$ |
| MLP on FashionMNIST | $88.54\% \pm 0.64\%$ | $85.12\% \pm 0.75\%$ | $\mathbf{89.13}\% \pm 0.86\%$ |
| CNN on SVHN | $95.35\% \pm 1.53\%$ | $94.53\% \pm 1.54\%$ | $\mathbf{96.45}\% \pm 1.04\%$ |
| CNN on CIFAR-10 | $69.34\% \pm 0.54\%$ | $70.84\% \pm 0.64\%$ | $\mathbf{72.54}\% \pm 0.93\%$ |
| AlexNet on CIFAR-10 | $\mathbf{75.64}\% \pm 0.64\%$ | $64.63\% \pm 1.55\%$ | $72.42\% \pm 0.53\%$ |

architectures (hidden dimension $\leq 64$). How-
ever, this difference is inverted in large networks: in the most extreme case, one weight update on
a network with 32 hidden layers and 1024 parameters per layer using iPC is 10 times faster than
that using BP. This is still below the result of Theorem 1 due to the large overhead introduced in our
implementation.

## 4 CLASSIFICATION EXPERIMENTS

We now demonstrate that iPC shows a similar level of generalization quality compared to BP. We
test the performance of iPC on different benchmarks. Since we focus on generalization quality in
this section, all methods are run until convergence, and we have used early stopping to pick the best
performing model. These experiments were performed using multi-batch training. In this case, we
lose our advantage in efficiency over BP, as we need to recompute the error every time a new batch
is presented. However, the proposed algorithm is still much faster than the original formulation of
PC, and yields a better classification performance.

**Setup of experiments:** We investigate image classification benchmarks using PC, iPC, and BP. We
first trained a fully connected network with 2 hidden layers and 64 hidden neurons per layer on the
MNIST dataset (LeCun & Cortes, 2010). Then, we trained a mid-size CNN with three convolu-
tional layers with $64 - 128 - 64$ kernels followed by two fully connected layers on FashionMNIST,
the Street View House Number (SVHN) dataset (Netzer et al., 2011), and CIFAR10 (Krizhevsky
et al., 2012) with no data augmentation. Finally, we trained AlexNet (Krizhevsky et al., 2012), a
large-scale CNN, on CIFAR10. To make sure that our results are not the consequence of a spe-
cific choice of hyperparameters, we performed a comprehensive grid-search on hyperparameters,
and reported the highest accuracy obtained. The search is further made robust by averaging over
5 seeds. Particularly, we tested over 8 learning rates (from 0.000001 to 0.01), 4 values of weight
decay $(0.0001, 0.001, 0.01, 0.1)$, and 3 values of the integration step $\gamma$ $(0.1, 0.5, 1.0)$, and each com-
bination of hyperparameters are evaluated with 5 seeds with mean and standard error reported. To
conclude, we have used no data augmentation in the experiments.

**Results:** In Table 1, iPC outperforms BP in all the small- and medium-size architectures. For the
simplest framework (MNIST on a small MLP), PC outperforms all the other training methods, with
iPC following by a tiny margin $(0.01\%)$. However, PC fails to scale to more complex problems,
where it gets outperformed by all the other training methods. The performance of iPC, on the other
hand, is stable under changes in size, architecture, and dataset. In fact, iPC reaches a slightly better
accuracy than BP on most of the considered tasks.

Table 2: Change of final accuracy when increasing the width.

| $C$ | 1 | 2 | 3 | 4 | 5 | 6 | 7 | 8 | 10 | 15 | 20 |
|---|---|---|---|---|---|---|---|---|---|---|---|
| BP | 67.92 | 71.23 | 71.65 | 72.64 | 73.35 | 73.71 | 74.19 | 74.51 | 74.62 | 75.08 | 75.51 |
| iPC | **70.61** | **74.12** | **74.91** | **75.88** | **76.61** | **77.04** | **77.48** | **77.41** | **76.51** | **76.55** | **76.12** |

**Change of width:** Table 1 shows that iPC performs better on a standard CNN than on AlexNet,
which has many more parameters and maxpooling layers. To investigate how iPC behaves when
adding max-pooling layers and increasing the width, we trained a CNN with three convolutional
layers $(8, 16, 8)$ and maxpools, followed by a fully connected layer (128 hidden neurons) on CI-
FAR10. We have also replicated the experiment by increasing the width of the network by multi-

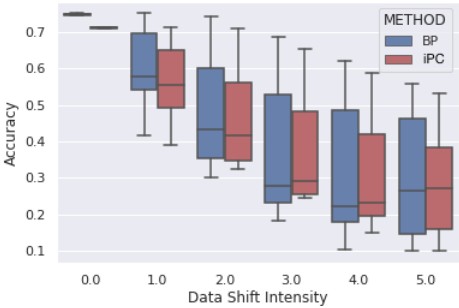 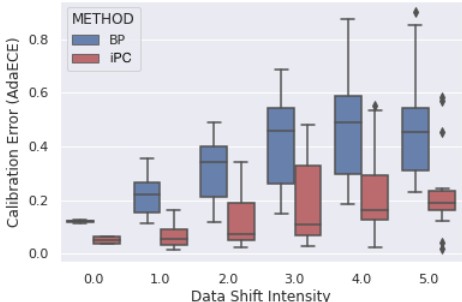

Figure 4: Robustness of BP and iPC under distribution shift (AlexNet on CIFAR10 under five different intensities of the corruptions rotation, Gaussian blur, Gaussian noise, hue, brightness, and contrast). *Left:* Comparable decline of model accuracy between BP and iPC. *Right:* iPC maintains model calibration significantly better than BP under distribution shift.

plying every hidden dimension by a constant $C$, (e.g., $C = 3$ means a network with 3 convolutional layers $(24, 48, 24)$, each followed by a maxpool, and a fully connected one $(384$ hidden neurons$)$). The results in Table 2 show that iPC (i) outperforms BP under each parametrization, (ii) needs less parameters to obtain good results, but (iii) sees its performance decrease, once it has reached a specific parametrization. This is in contrast to BP, which is able to generalize well even when extremely overparametrized. This suggests that iPC is more efficient than BP in terms of the number of parameters, but that finding the best parameters for iPC may need some extra tuning.

## 4.1 ROBUSTNESS AND CALIBRATION

Robustness and uncertainty quantification in deep learning have become a topic of increasing interest in recent years. While neural networks trained via BP reach a strong model performance, their lack of explainability and robustness has been widely studied (Abdar et al., 2021; Ovadia et al., 2019). Recently, it has been noted that treating classifiers as generative energy-based models benefits the robustness of the model (Grathwohl et al., 2019). As PC is precisely an energy-based classifier, originally developed for generation tasks, we postulate that iPC possesses better robustness and calibration characteristics than BP. Calibration describes the degree to which predicted logits matches the empirical distribution of observations given the prediction confidence. One may use a calibrated model's output to quantify the uncertainty in its predictions and interpret it as probability—not just model confidence. Let $\hat{P}$ be our random prediction vector indicating the model's confidence that the prediction $\hat{Y}$ is correct. We say $\hat{P}$ is well-calibrated, if the model confidence matches the model performance, i.e., $\mathbb{P}(\hat{Y} = Y | \hat{P} = p) = p$ (Guo et al., 2017). We measure the deviation from calibration using the adaptive expected calibration error (AdaECE), which estimates $\mathbb{E}[|\mathbb{P}(\hat{Y} = Y | \hat{P} = p) - p|]$ (Nguyen & O'Connor, 2015). In recent years, it has become well-known that neural networks trained with BP tend to be overconfident in their predictions (Guo et al., 2017) and that miscalibration increases dramatically under distribution shift (Ovadia et al., 2019). More details on the experiments are in the supplementary material.

**Results:** Our results are shown in Fig. 4. The boxplots indicate the distributions of accuracy (left) and calibration error (right) over various forms of data corruption with equal levels of intensity. We find that the discriminative performance of the BP and iPC models are comparable under distribution shift. Both models keep a reasonable classification performance for mild corruptions, but show accuracies going down to chance performance under extreme corruptions. The calibration of model output, however, differs strongly: The iPC-trained model yields better calibrated outputs and is able to signal its confidence a lot better. This is essential for using the model output as indication of uncertainty. On in-distribution data, we observe that iPC yields an average calibration error of $0.05$, whereas BP yields $0.12$. Moreover, we observe that the increase in calibration error is a lot weaker for iPC: The median calibration error of the iPC model is lower across all levels of shift intensities compared to that of BP for the mildest corruption. Furthermore, iPC displays better calibration up to level 3 shifts than BP does on in-distribution data. This has potentially a strong impact of applying either method in safety-critical applications.

## 5   RELATED WORKS

Several previous research efforts aim to achieve supervised learning in a biologically plausible way. One is to explore how the error can be encoded differently than in BP where the error is not encoded locally. One of the earliest works was to use a second set of "error" neurons that can act as the feedback variables (encoding error in BP) (Stork, 1989; Schwartz, 1993). Another promising assumption is that the error can be represented in neurons' dendrites (Körding & König, 2001; 2000; Richards & Lillicrap, 2019; Sacramento et al., 2018). Such efforts are unified in (Lillicrap et al., 2020), with a broad range of works (Pineda, 1987; 1988; O'Reilly, 1996; Ackley et al., 1985; Hinton et al., 1995; Bengio, 2014; Lee et al., 2015) encoding the error term in activity differences.

Neuroscience-inspired algorithms have recently gained the attention of the machine learning community, due to interesting properties such as locality, autonomy and their energy-based formulation. To this end, multiple works have used PC to tackle machine learning problems, from generation tasks (Ororbia & Kifer, 2020), to image classification on complex datasets such as ImageNet (He et al., 2016), associative memories (Salvatori et al., 2021), continual learning (Ororbia et al., 2020), and NLP (Pinchetti et al., 2022). There is a more theoretical line of work that is related to the free energy principle and active inference (Friston, 2008; 2010; Friston et al., 2006; 2016), which aims to model learning, perception, and behavior as an imperative to minimize a free energy. While being initially a theoretical framework, it has been used in multiple applications in fields such as control theory (Baltieri & Buckley, 2019; Friston, 2011) and reinforcement learning (Friston et al., 2009). To conclude, it is important to note that iEM is not the only formulation that improves the efficiency of the original EM, as some other variations have been proposed, such as an online version (Cappé & Moulines, 2009), a stochastic one (Chen et al., 2018), or a newer incremental version (Karimi et al., 2019) inspired by the SAGA algorithm (Defazio et al., 2014).

## 6   DISCUSSION

In this paper, we have proposed a biologically inspired learning rule, called *incremental predictive coding* (*iPC*) motivated by the *incremental EM* algorithm. iPC enables all the computations to be executed *simultaneously*, *locally*, and *autonomously*, and has theoretical convergence guarantees in non-asymptotic time (Karimi et al., 2019). This allows a solid gain in efficiency compared to the original formulation of PC as well as BP in the full-batch case, as shown with extensive experiments, with no drawbacks in the converging to a minimum of the loss. This is confirmed by the good experimental results in terms of accuracy and robustness in classification tasks.

An interesting aspect worth discussing, is the time step that triggers the weight update in the three variations of PC: the original formulation, Z-IL, and, now, iPC. The first method updates the parameters only in the last step of the inference, when the neural activities have converged. This has interesting theoretical properties, as it has been shown to simulate how learning is performed in multiple models of cortical circuits, as its credit assignment converges to an equilibrium called *prospective configuration* (Song et al., 2022). The second, Z-IL, shows that it suffices to time the updates at different levels of the hierarchy in different moments of the inference, to exactly replicate the update given by BP on any possible neural network (Song et al., 2020; Salvatori et al., 2022b). This is interesting, as it connects PC, a theory developed to model credit assignment in the brain, to BP, a method developed to train deep learning models. Our newly proposed iPC, on the other hand, updates the parameters continuously, resulting in great gains in terms of efficiency, and no apparent loss in terms of performance. Future work will investigate whether there are better variations of iPC, or whether the optimal update rule can be learned with respect to specific tasks and datasets. Again, the answer may lie in some variations of the EM algorithm, such as *dynamical* EM (Anil Meera & Wisse, 2021; Friston et al., 2008), or in an implementation of precision-weighted prediction errors, as in (Jiang & Rao, 2022).

On a broader level, this work shrinks the gap between computational neuroscience and machine intelligence by tackling the problem of the computational efficiency of neuroscience-inspired training algorithms. Advances in this direction are also interesting from the perspective of hardware implementations of deep learning on energy-based chips, such as analog and quantum computers. In this case, iPC is an interesting improvement, as it is still not known how external control can be implemented on these chips, and hence algorithms able to train neural networks in a fully automatic fashion may play an important role in the future.

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

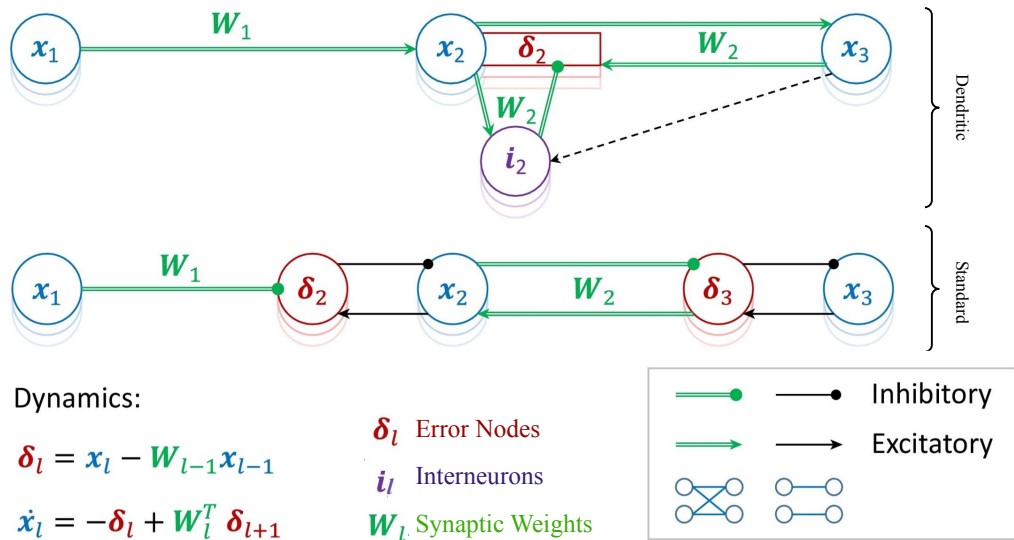

Figure 5: Standard and dendritic neural implementation of predictive coding. The dendritic implementation makes use of interneurons $i_l = W_l x_l$ (according to the notation used in the figure). Both implementations have the same equations for all the updates, and are thus equivalent; however, dendrites allow a neural implementation that does not take error nodes into account, improving the biological plausibility of the model. Figure taken and adapted from (Whittington & Bogacz, 2019).

## A  A DISCUSSION ON BIOLOGICAL PLAUSIBILITY

In this section, we discuss the biological plausibility of the proposed algorithm, a topic overlooked in the main body of this paper. In the literature, there is often a disagreement on whether a specific algorithm is biologically plausible or not. Generally, it is assumed that an algorithm is biologically plausible when it satisfies a list of properties that are also satisfied in the brain. Different works consider different properties. In our case, we consider as list of minimal properties that include local computations and lack of a global control signals to trigger the operations. Normally, predictive coding networks take error nodes into account, often considered implausible from the biological perspective (Sacramento et al., 2018). Even so, the biological plausibility of our model is not affected by this: it is in fact possible to map PC on a different neural architecture, in which errors are encoded in apical dendrites rather than separate neurons (Sacramento et al., 2018; Whittington & Bogacz, 2019). Graphical representations of the differences between the two implementations can be found in Fig. 5, taken (and adapted) from (Whittington & Bogacz, 2019). Furthermore, our formulation is more plausible than the original formulation of PC, as it is able to learn without the need of external control signals that trigger the weight update.

# B    PSEUDOCODES OF Z-IL AND PC

---

**Algorithm 2** Learning a dataset $\mathcal{D} = \{y_i\}$ with PC.

---

1: **Require:** For every $i$, $x_i^{(0)}$ is fixed to $y_i$,
2: **for** $t = 0$ to $T$ **do**
3:    For every $i$ and $l$, update $x^{(l)}$ to minimize $F$ via Eq.(7)
4:    **if** t = T **then**
         For every $l$ update each $\theta^{(l)}$ to minimize $F$ via Eq. (8)
5:    **end if**
6: **end for**

---

---

**Algorithm 3** Learning one training pair $(s^{\text{in}}, s^{\text{out}})$ with Z-IL

---

1: **Require:** $x_0^L$ is fixed to $s^{\text{in}}$, $x_0^0$ is fixed to $s^{\text{out}}$.
2: **Require:** $x^{(l)} = \mu^{(l)}$ for $l \in \{1, ..., L-1\}$, and $t = 0$.
3: **for** $t = 0$ to $T$ **do**
4:    **for** each level $l$ **do**
5:        Update $x^{(l)}$ to minimize $F$ via Eq.(7)
6:        **if** $t = l$ **then**
7:            Update $\theta^{(l)}$ to minimize $F$ via Eq.(8).
8:        **end if**
9:    **end for**
10: **end for**

---

Table 3: Theoretical Efficiency of PC, Z-IL, BP, and iPC.

| | One inference step | PC | Z-IL | BP | iPC |
|---|---|---|---|---|---|
| Number of MMs per weight update | $(2L-1)$ | $(2L-1)T$ | $(2L-1)(L-1)$ | $(2L-1)$ | $(2L-1)$ |
| Number of SMMs per weight update | 2 | $2T$ | $2(L-1)$ | $(2L-1)$ | 2 |

## C  ON THE EFFICIENCY OF PC, BP, AND iPC

In this section, we discuss the time complexity and efficiency of PC, BP, Z-IL, and iPC. We now start with the first three, and introduce a metric that we use to compute such complexity. This metric is the number of *simultaneous matrix multiplications* (*SMMs*), i.e., the number of non-parallelizable matrix multiplications needed to perform a single weight update. It is a reasonable approximation of running time, as multiplications are by far the most complex operation ($\approx \mathcal{O}(N^3)$) performed by the algorithm.

### C.1  COMPLEXITY OF PC, BP, AND Z-IL

**Serial Complexity:** To complete a single update of all weights, PC and Z-IL run for $T$ and $(L-1)$ inference steps, respectively. To study the complexity of the inference steps we consider the number of *matrix multiplications* (*MMs*) required for each algorithm: One inference step requires $(2L-1)$ MMs: $L$ for updating all the errors, and $(L-1)$ for updating all the value nodes (Eq. equation 6). Thus, to complete one weight update, PC and Z-IL require $(2L-1)T$ and $(2L-1)(L-1)$ MMs, respectively. Note also that BP requires $(2L-1)$ MMs to complete a single weight update: $L$ for the forward, and $(L-1)$ for the backward pass. These numbers are summarized in the first row of Table 3. According to this measure, BP is the most efficient algorithm, Z-IL ranks second, and PC third, as in practice $T$ is much larger than $L$. However, this measure only considers the total number of matrix multiplications needed, without considering whether some of them can be performed in parallel, which could significantly reduce the time complexity. We now address this problem.

**Parallel complexity:** The MMs performed during inference can be parallelized across layers. In fact, computations in Eq. equation 6 are layer-wise independent, thus $L$ MMs that update all the error nodes take the time of only one MM if properly parallelized. Similarly, in Eq. equation 6, $(L-1)$ MMs that update all the value nodes take the time of only one MM if properly parallelized. As a result, one inference step only takes the time of 2 MMs if properly parallelized (since, as stated, it consists of updating all errors and values via Eq. equation 6). Thus, one inference step takes 2 SMMs; one weight update with PC and Z-IL takes $2T$ and $2(L-1)$ SMMs, respectively. Since no MM can be parallelized in BP (the forward pass in the network and the backward pass of error are both layer-dependent), before performing a single weight update, $(2L-1)$ SMMs are required. These numbers are summarized in the second row of Table 3. Overall, measured over SMMs, BP and Z-IL are equally efficient (up to a constant factor), and faster than PC.

### C.2  COMPLEXITY OF iPC

To complete one weight update, iPC requires one inference step, thus $(2L-1)$ MMs or 2 SMMs, as also demonstrated in the last column of Table 3. Compared to BP, iPC takes around $L$ times less SMMs per weight update, and should hence be significantly faster in deep networks. Intuitively, this is because matrix multiplications in BP have to be done sequentially along layers, while the ones in iPC can all be done in parallel across layers (Fig. 6). More formally, we have the following theorem, which holds when performing full-batch training:

**Theorem 1.** *Let $M$ and $M'$ be two equivalent networks with $L$ layers trained on the same dataset. Let $M$ (resp., $M'$) be trained using BP (resp., iPC). Then, the time complexity measured by SMMs needed to perform one full update of the weights is $\mathcal{O}(1)$ and $\mathcal{O}(L)$ for iPC and BP, respectively.*

*Proof.* Consider training on an MLP with $L$ layers, and update weights for multiple times on a single datapoint. Generalizations to multiple datapoints and multiple mini-batches are similar and will be provided after. We first write the equations needed to be computed for iPC to produce one weight

update:

$$x_{i,t}^{(L)} = s_i^{in} \text{ and } x_{i,t}^{(0)} = s_i^{out}$$

$$\hat{x}_{i,t}^{(l)} = \sum_{j=1}^{n^{l-1}} \theta_{i,j}^{(l+1)} f(x_{j,t}^{(l+1)}) \qquad \text{for } l \in \{1, \dots, L\} \tag{9}$$

$$\varepsilon_{i,t}^{(l)} = x_{i,t}^{(l)} - \hat{x}_{i,t}^{(l)} \qquad \text{for } l \in \{1, \dots, L\}$$

$$x_{i,t+1}^{(l)} = x_{i,t}^{(l)} + \gamma \cdot \left( -\varepsilon_{i,t}^{(l)} + x_{i,t}^{(l)} \sum_{k=1}^{n^{(l+1)}} \varepsilon_{k,t}^{(l+1)} \theta_{k,i}^{(l)} \right) \qquad \text{for } l \in \{1, \dots, L\} \tag{10}$$

$$\theta_{i,j,t+1}^{(l)} = \theta_{i,j,t}^{(l)} - \alpha \cdot \varepsilon_{i,t}^{(l+1)} f(x_{j,t}^{(l)}) \qquad \text{for } l \in \{1, \dots, L\}. \tag{11}$$

We then write the three equations needed to be computed for BP to produce one weight update:

$$x_{i,t}^{0} = s_i^{in}$$

$$x_{i,t}^{(l)} = \sum_{j=1}^{n^{l-1}} \theta_{i,j}^{(l+1)} f(x_{j,t}^{(l+1)}) \text{ for } l \in \{1, \dots, L\} \tag{12}$$

$$\varepsilon_{i,t}^{(L)} = s_i^{out} - x_{i,t}^{(L)}$$

$$\varepsilon_{i,t}^{(l)} = f'\left(x_{i,t}^{(l)}\right) \sum_{k=1}^{n^{(l+1)}} \varepsilon_{k,t}^{(l+1)} \theta_{k,i}^{(l)} \text{ for } l \in \{L, \dots, 1\} \tag{13}$$

$$\theta_{i,j,t+1}^{(l)} = \theta_{i,j,t}^{(l)} - \alpha \cdot \varepsilon_{i,t}^{(l+1)} f(x_{j,t}^{(l)}) \text{ for } l \in \{1, \dots, L\}.$$

First, we notice that the matrix multiplication (MM) is the most complex operation. Specifically, for two adjacent layers with the size of $n^l$ and $n^l$, the complexity of MM is $\mathcal{O}(n^l n^l)$, but the maximal complexity of the other operations is $\mathcal{O}(\max n^l, n^l)$. In the above equations, only equations with MM are numbered, which are the equations that we investigate in our complexity analysis.

Eq. equation 9 for iPC takes $L$ MMs, but one SMM, since the the for-loop for $l \in \{1, \dots, L\}$ can run in parallel for different $l$. This is further because the variables on the right side of Eq. equation 9 are immediately available. Differently, Eq. equation 12 for iPC takes $L$ MMs, and also $L$ SMMs, since the for-loop for $l \in \{1, \dots, L\}$ has to be executed one after another, following the specified order $\{2, \dots, L\}$. This is further because the qualities on the right side of Eq. equation 12 are immediately available, but require to solve Eq. equation 12 again for another layer. That is, to get $x_{i,t}^{(L)}$, Eq. equation 12 has to be solved recursively from $l = 1$ to $l = L$.

Similar sense applies to the comparison between Eqs. equation 10 and equation 13. Eq. equation 10 for iPC takes $L - 1$ MMs but 1 SMMs; Eq. equation 13 for BP takes $L - 1$ MMs and also $L - 1$ SMMs.

Overall, Eqs. equation 9 and equation 10 for iPC take $2L - 1$ MMs but 2 SMMs; Eqs. equation 12 and equation 13 for BP take $2L - 1$ MMs and also $2L - 1$ SMMs. Then, the time complexity measured by SMMs needed to perform one full update of the weights is $\mathcal{O}(1)$ and $\mathcal{O}(L)$ for iPC and BP, respectively.

### C.3 EFFICIENCY ON ONE DATA POINT

To make the difference more visible and provide more insights, we explain this in detail with a sketch of this process on a small network in Fig. 6, where the horizontal axis of $m$ is the time step measured by simultaneous matrix multiplications (SMMs), i.e., within a single $m$, one can perform one matrix multiplication or multiple ones in parallel; if two matrix multiplications have to be executed in order (e.g., the second needs results from the first), they will need to be put into

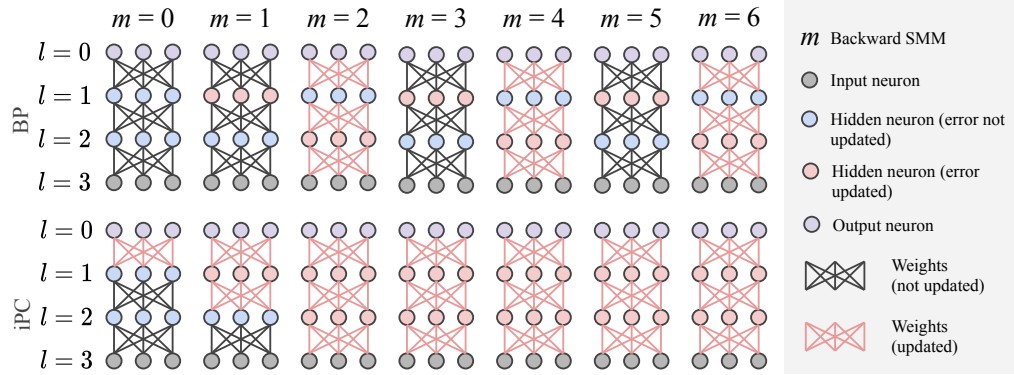

Figure 6: Graphical PClustration of the efficiency over backward SMMs of BP and iPC on a 3-layer network. iPC never clears the error (red neurons), while BP clears it after every update. This allows iPC to perform 5 full and 2 partial updates of the weights in the first 6 SMMs. In the same time frame, BP only performs 3 full updates. Note that the SMMs of forward passes are excluded for simplicity, w.l.o.g., as the insight from this example generalizes to the SMMs of the forward pass.

two steps of $m$. Note that we only consider the matrix multiplications for the backward pass, i.e., the matrix multiplications that backpropagate the error of a layer from an adjacent layer for BP and the inference of Eq. equation 6 for iPC, thus the horizontal axis $m$ is strictly speaking "Backward SMM". The insight for the forward pass is similar as that of the backward pass. As it has been said, for BP, backpropagating the error from one layer to an adjacent layer requires one matrix multiplication; for iPC, one step of inference on one layer via Eq. equation 6 requires one matrix multiplication. BP and iPC are presented in the first and second rows, respectively. Before both methods are able to update weights in all layers, they need two matrix multiplications for spreading the error through the network, i.e., a weights update of all layers occurs for the first time at $m = 2$ for both methods. After $m = 2$, BP cleared all errors on all neurons, so at $m = 3$, BP backpropagates the error from $l = 0$ to $l = 1$, and at $m = 4$, BP backpropagates the error from $l = 1$ to $l = 2$ after which it can make an update of weights at all layers again for the second time. Note that the matrix multiplication that backpropagates errors from $l = 1$ to $l = 2$ at $m = 4$ cannot be put at $m = 3$, as it requires the results of the matrix multiplication at $m = 3$, i.e., it requires the error to be backpropagated to $l = 1$ from $l = 0$ at $m = 3$. However, this is different for iPC. After $m = 2$, iPC does not reset $x_{i,t}^l$ to $\mu_{i,t}^l$, i.e., the error signals are still held in $\varepsilon_{i,t}^l$. At $m = 3$, iPC performs two matrix multiplications in parallel, corresponding to two inferences steps at two layers, $l = 1$ and $l = 2$, updating $x_{i,t}^l$, and hence the error signals are held in $\varepsilon_{i,t}^l$ of these two layers. Note that the above two matrix multiplications of two inference steps can run in parallel and be put into a single $m$, as inference requires only locally and immediately available information. In this way, a weight update in iPC is able to be performed at every $m$ ever since the very first few steps of $m$.

## D    TRAINING DETAILS

We now list some additional details to reproduce our results.

### D.1    EXPERIMENTS OF EFFICIENCY

The experiments for the efficiency of generative models were run on fully connected networks with 128, 256 or 512 hidden neurons, and $L \in \{4, 5\}$. Every network was trained on CIFAR10 or Tiny Imagenet with learning rates $\alpha = 0.00005$ and $\gamma = 0.5$, and $T \in \{8, 12, 16\}$. The experiments on discriminative models are performed using networks with 64 hidden neurons, depth $L \in \{3, 4, 6\}$, and learning rates $\alpha = 0.0001$ and $\gamma = 0.5$. The networks trained with BP have the same learning rate $\alpha$. All the plots for every combination of hyperparameters can be found in Figures 8 and 7.

## D.2 Experiments of Generalization Quality

As already stated in the paper body, to make sure that our results are not the consequence of a specific choice of hyperparameters, we performed a comprehensive grid search on hyperparameters, and reported the highest accuracy obtained, and the search is further made robust by averaging over $5$ seeds. Particularly, we tested over $8$ learning rates (from $0.000001$ to $0.01$), $4$ values of weight decay $(0.0001, 0.001, 0.01, 0.1)$, and $3$ values of the integration step $\gamma$ $(0.1, 0.5, 1.0)$. We additionally verified that the optimized value of each hyperparameter lies within the searched range of that hyperparameter. As for additional details, we used standard Pytorch initialization for the parameters. For the hardware, we used a single Nvidia GeForce RTX 2080 GPU on an internal cluster. Despite the large search, most of of the best results were obtained using the following hyperparameters: $\gamma = 0.5$ ($\gamma = 1$ for Alexnet), $\alpha = 0.00005$.

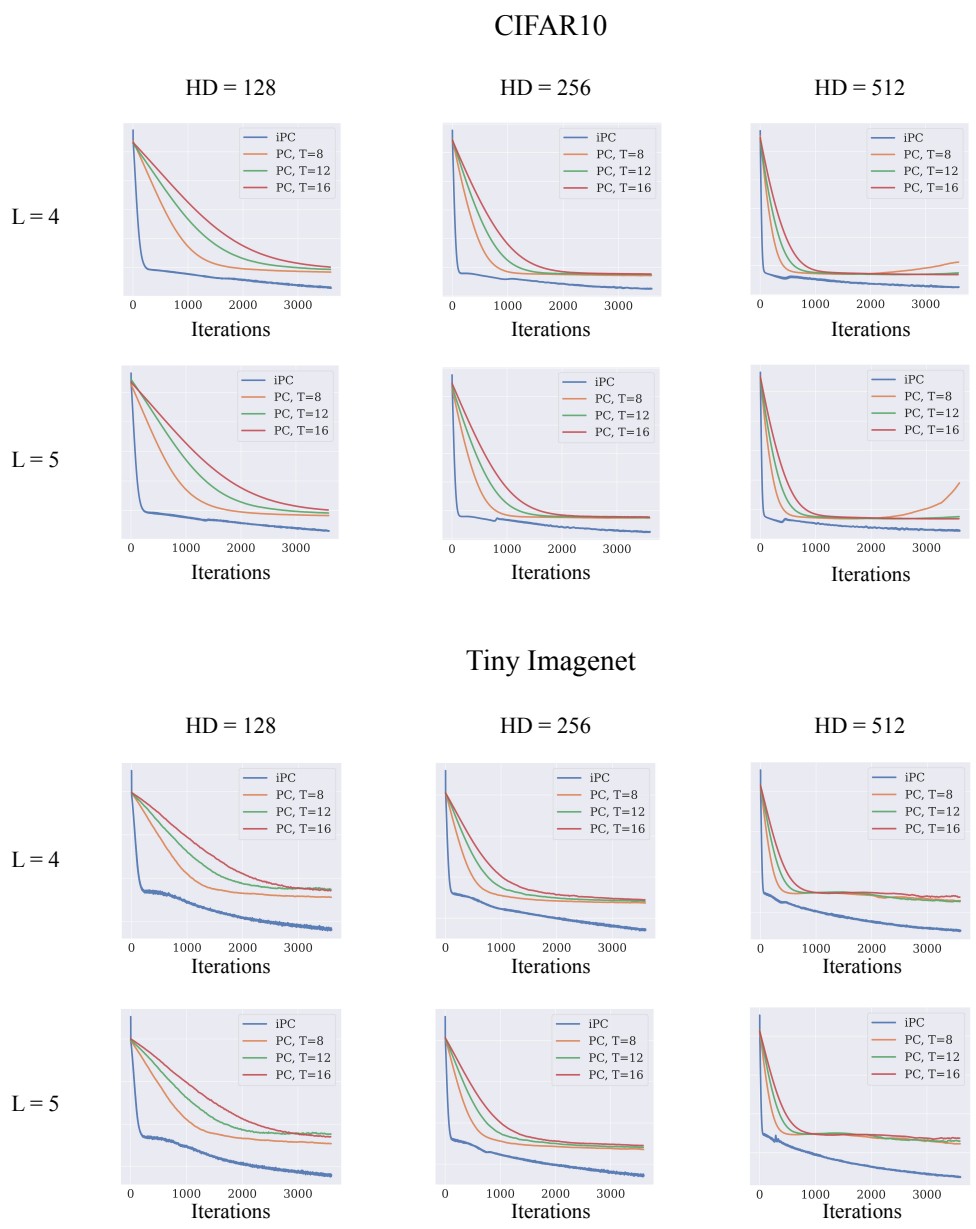

Figure 7: Efficiency of multiple generative networks trained with PC.

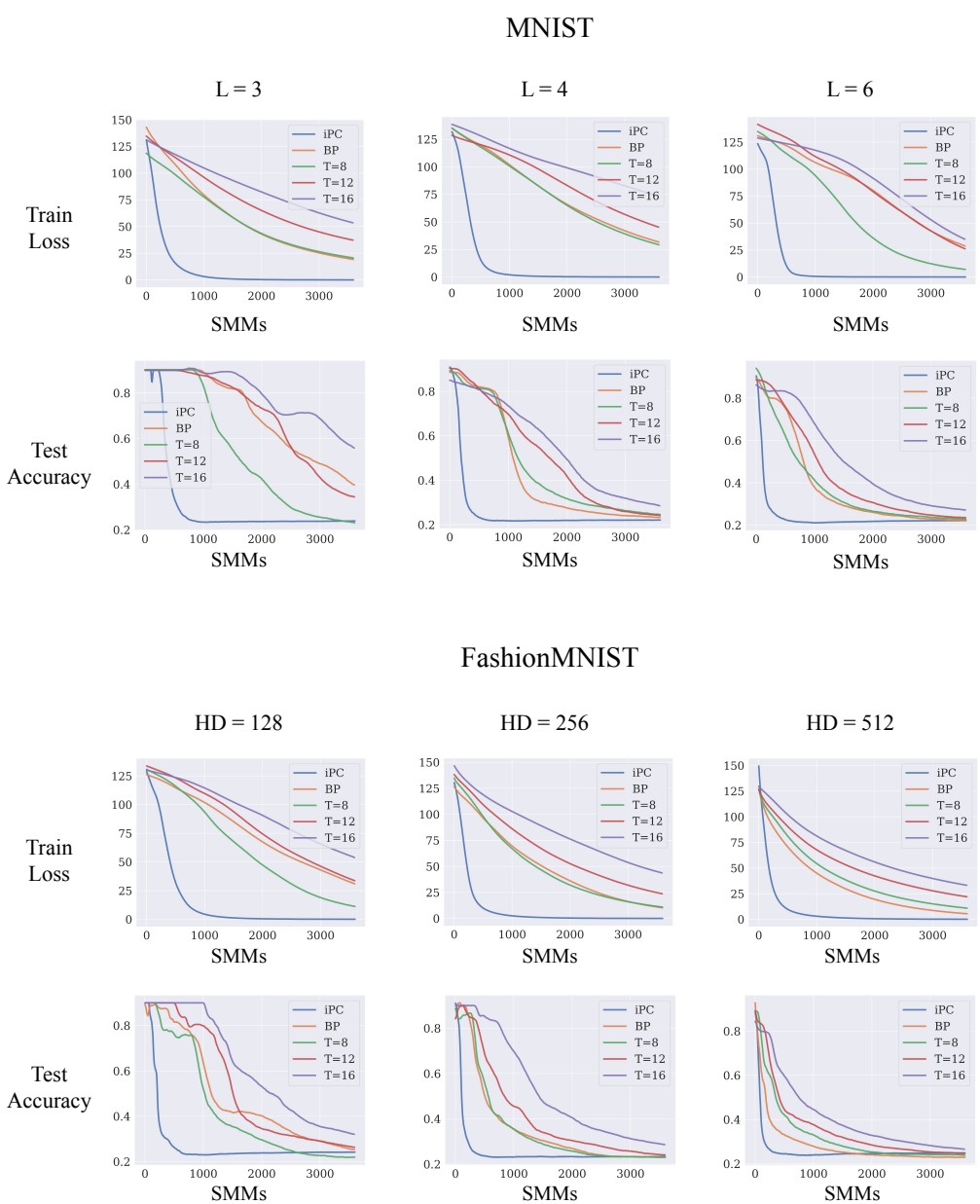

Figure 8: Efficiency of multiple discriminative networks trained with PC and BP.

