# OpenReview forum: "Incremental Predictive Coding: A Parallel and Fully Automatic Learning Algorithm"
_ICLR.cc/2023/Conference — Submitted to ICLR 2023_

### Official Review · Reviewer_szhT · 2022-10-23

**Confidence:** 4
**Clarity, Quality, Novelty And Reproducibility:** The paper is very clearly written. Bu…
**Correctness:** 4
**Technical Novelty And Significance:** 1
**Empirical Novelty And Significance:** 2
**Recommendation:** 3

**Strength And Weaknesses:**

Strength: Empirical results show faster convergence
Weekness: There is nothing novel in this paper. Neither does this paper claim to introduce the prospective configuration algo (I am guessing it is done in another submission), nor does it need to show convergence (which follows from Neal and Hinton). It is therefore just an implementation paper. Had the experiments been large and comprehensive, there would be reason to consider this paper.

**Summary Of The Paper:**

This paper is basically an implementation of the prospective configuration algoithm (Yuhang Song, et al, Inferring neural activity before plasticity: A foundation for learning beyond backpropagation) where each layer of a heirarchical Gaussian generative model (with the further assumption that the covariance matrix of the gaussians is identity), is updated in parallel. Convergence guarantees to some local optima come from Neal and Hinton's incremental EM paper.

**Summary Of The Review:**

The paper has very limited novelty. It is just an implementation of ideas found in other papers.

---

> ### Author Response · Authors · 2022-11-11
> **Answer to Reviewer**
>
> We thank the author for the feedback on our work.
> >  This paper is basically an implementation of the prospective configuration algoithm (Yuhang Song, et al, Inferring neural activity before plasticity: A foundation for learning beyond backpropagation) where each layer of a heirarchical Gaussian generative model (with the further assumption that the covariance matrix of the gaussians is identity), is updated in parallel.
>
> The statement that our work is an implementation of the prospective configuration algorithm is incorrect.  The paper that you cite uses the original implementation of predictive coding, and does NOT update the weights in parallel with the neural activities: it first updates the value nodes until convergence, and THEN performs a weight update. This is shown in Figs.1b and 2c of the paper you cite:
>
>  https://www.biorxiv.org/content/10.1101/2022.05.17.492325v2.full.pdf
>
> In detail, the description of Fig. 1c states the following:
>
> “In prospective configuration, neural activity settles into a new configuration (purple of different intensity) before weight modification (left).”
>
> This clearly states that FIRST the neural activities converge, and THEN a weight update is performed. Something similar is stated in the description of Fig. 2b:
>
> “Here, it is revealed that the relaxation infers the prospective neural activity, towards which the weights are THEN modified”
>
>
> Again, this means that FIRST the neural activities converge, and THEN a weight update is performed. The implementation details of this work further confirm that the algorithm of the paper that you cite is NOT equivalent to iPC. In fact, the “Methods” of the paper actually provide the actual equations, and the pseudocode of the algorithm. Particularly, we refer to Eq. (4). Right above Eq. (4), the authors state the following (line 770):
>
> “Then, relaxation is run until convergence, after which the weights are updated using the activity at convergence to further decrease the energy.”
>
> Again, this is also visible in the pseudocode that the authors provide on page 26. Here, the value nodes are updated until convergence (lines 6-13). The weights are only updated at lines 15-17 when the relaxation process is complete.  Hence, our work is NOT an implementation of the prospective configuration algorithm, as that is a theoretical study of the properties of predictive coding, while our work is the proposal of a faster and better optimization techniques for predictive coding networks. All in all, there is absolutely NO intersection in the contributions of our work and the one you have cited, as each layer of a hierarchical Gaussian generative model  is NOT updated in parallel there.
>
> On the weaknesses:
>
> > Neither does this paper claim to introduce the prospective configuration algorithm…
>
> Again, the prospective configuration algorithm is an independent line of work, which we cannot claim, as it is equivalent to the predictive coding algorithm introduced by Rao and Ballard from 1999. In our work, we propose an improvement over that update rule, which may benefit researchers working in this field. There is absolutely no intersection between the contributions of the paper that you have cited and ours.
>
> > nor does it need to show convergence (which follows from Neal and Hinton).
>
> As already stated in the paragraph about novelty that we have provided, this is actually an advantage of the method that we have proposed, and not a limitation:  When an algorithm is based on a well-studied mathematical field, it is often possible to derive specific results ‘for free’, as they are probably derivations of already well-studied problems. The fact that we can use the theory of statistical inference for free to prove results is a strength of predictive coding (which is a well-worth result on its own), used in this work to provide a faster and better performing algorithm with theoretical guarantees.

---

> > ### Comment · Reviewer_szhT · 2022-11-22
> > **Comment to author comment**
> >
> > As the authors themselves point out, the difference is between EM and incremental EM. In one case you first let the neural activity settle and then you update the weights, and in the other you do them together. This I find is too small a contribution (noting further that this has been known in the EM/incremental EM liturature).

---

### Official Review · Reviewer_fv42 · 2022-10-26

**Confidence:** 3
**Correctness:** 3
**Technical Novelty And Significance:** 3
**Empirical Novelty And Significance:** 3
**Recommendation:** 5

**Clarity, Quality, Novelty And Reproducibility:**

> [main limitation of predictive coding being] "lack of efficiency"

I'm not sure about this. I think it's more like: predictive coding models are harder to train and do not work as well as feedforward alternatives.

> "can match backpropagation in its most important property: generalization capabilities"

I'm not sure about this. I think the the common view is that the most important property of backpropagation is that it can manage credit assignment across long chains of modules/neurons, and enable learning of large nonlinear systems.

Notation-wise, it seems like almost every variable in the paper has a bar over it. Maybe all bars can be removed, to make things simpler.

In Eq. 1, ybar is defined as "the generated vector", but also p(ybar, \thetabar) is described as "likelihood of the data given the causes", suggesting ybar is actually the data. Can this be clarified please?

Some citation issues -- parenthetical and non-parenthetical citations are mixed up, making  it hard to read. (e.g., "intractable Friston (2003)", and "Following (Rao ...),".

optimizingthe -> optimizing the
gradient descend -> gradient descent
 – the slow inference phase –there  -> ---the slow inference phase---there

> "Note that this increased speed does not harm the final performance, as the iEM algorithm has been proven to converge to a stationary point of the loss function, the same as EM or backpropagation"

This is not very convincing. I think what we really want to know is whether or not this works in practice. I think even incremental EM (without the predictive coding interpretation) is typically less stable than EM.

> "This algorithm is able to obtain performance competitive to BP on image classification
tasks."

This is a problematic claim. There are many algorithmic components involved in applying BP on image classification tasks. I don't think such a general claim can be made accurately.

> "we provide enough evidence that the speed up against BP in full batch training is theoretically possible using iPC"

I'm not sure -- what makes it "enough"?

> "the concept of inference and iterations are not present in BP"

I'm not sure what is meant by this. These concepts are certainly present in BP-based models. Why not simply count the number of gradient steps?

> "we first prove this formally using the number of non-parallel matrix multiplications needed to perform a weight update as a metric"

I don't know what to do with this "proof". Is this useful? What exactly is meant by "full batch" anyway? Does this mean putting the whole dataset into memory? To me this seems totally impractical.

> "This is still below the result of Theorem 1 due to the large overhead introduced in our implementation."

OK but is there any feasible way forward on this? To me it seems more like the theorem is not very useful. Also, I would like to know: What are the actual runtimes here? I understand the ratio is somewhat helpful for a comparison, but it obscures the actual values in play here.

The descriptions in "setup of experiments" and "change of width" were a bit confusing. Why is everything sequential ("first we trained this, then we trained that")? Why does the order matter? Are you training the model, then adding more parameters, then continuing training?  (I expect not...)

Instead of Theorem 1 and the proof and so on, why not simply count the actual number of matrix multiplications?

I am not sure about the comparison between BP and iPC in the experiments. Is the network architecture exactly the same? Normally with BP we have a feedforward architecture, without a generative interpretation.

Sometimes the term "Z-IL" appears but I think it was never defined. It appears more in the discussion. What is this?

**Strength And Weaknesses:**

I have some issues with the claims and descriptions made in this paper. I want to acknowledge early that some of my issues may be misunderstandings, because I am still learning about predictive coding here.

The high-level issue for me is: the background seems to be "predictive coding vs backpropagation", where predictive coding is essentially equivalent to expectation-maximization, but this is not the way I think about predictive coding. This sentence matches my understanding better: "PC is based on the assumption that brains implement an internal generative model of the world, needed to predict incoming stimuli (or data)." Expectation-maximization is one way to optimize a predictive coding model, and backpropagation is another, and there are more still, like Hebbian learning -- these are all optimization techniques, and choices here are orthogonal to the issue of predictive inference vs non-predictive inference.


**Summary Of The Paper:**

This paper points out that the predictive coding implementation from Rao and Ballard is essentially expectation maximization, and therefore can be sped up by using the incremental (aka partial) variant of EM instead of the full version. The paper also argues that this is more efficient than backpropagation, at least for large networks and full-batch learning, and has better uncertainty calibration than models trained with backprop.


**Summary Of The Review:**

I enjoyed reading the paper and I think the topic is important, and a great fit for ICLR. I pointed out a variety of claims that appear questionable to me, which maybe stem from thinking about predictive coding as a model formulation (as I do, which I think is standard) vs. thinking about predictive coding as an optimization technique (as the paper does).

---

> ### Author Response · Authors · 2022-11-11
> **Answer to the reviewer**
>
> We thank the reviewer for the detailed feedback. We have addressed all the sentences pointed out by the reviewer in the updated version of the manuscript.
>
>
>
> > What does it mean that the concepts of steps and iterations are not present in BP? These concepts are certainly present in BP-based models. Why not simply count the number of gradient steps?
>
> The problem with counting the number of gradient steps is that a single gradient step performed by BP uses different operations than a step of PC, with a complexity that scales linearly with the number of layers. This is not the case with PCs, where the complexity is constant with respect to the number of layers, as every operation can be performed in parallel. Because of this, comparing PC and BP in terms of “steps” would be wrong. Hence, we needed a fair metric to compare the complexity of the two models, which is the number of simultaneous matrix multiplications. We have better specified this in the updated manuscript.
>
> > "we first prove this formally using the number of non-parallel matrix multiplications needed to perform a weight update as a metric"I don't know what to do with this "proof". Is this useful? What exactly is meant by "full batch" anyway? Does this mean putting the whole dataset into memory?
>
> It is useful, as it shows the much improved efficiency of iPC over the original formulation of PC, and also an advantage over BP in a very restricted case: full batch training. We agree that it is mostly impractical, as most of large-scale applications would not fit a whole dataset into memory. However, we still believe it is an interesting result and hence included it in the paper, as it could be useful in cases where we deal with small datasets of low-dimensional datapoints, such as continual learning, and we need to quickly memorize them before being faced with the next data.
>
>
>
> > "This is still below the result of Theorem 1 due to the large overhead introduced in our implementation."
> OK but is there any feasible way forward on this? To me it seems more like the theorem is not very useful. Also, I would like to know: What are the actual runtimes here? I understand the ratio is somewhat helpful for a comparison, but it obscures the actual values in play here.
>
>
> Thank you for the suggestion, we have provided a table with the actual runtimes of every experiment in the following reply. However, note that the actual runtimes are not interesting per se, as every experiment was run on a CPU.
>
>
> > The descriptions in "setup of experiments" and "change of width" were a bit confusing. Why is everything sequential ("first we trained this, then we trained that")? Why does the order matter? Are you training the model, then adding more parameters, then continuing training? (I expect not...)
>
> Thank you for the pointer, we have reworded the paragraph. (The order does not matter, and we do not add more parameters during training!)
>
>
> > Instead of Theorem 1 and the proof and so on, why not simply count the actual number of matrix multiplications?
>
> The proof is basically obtained by counting the number of non-parallel multiplications. We have decided to summarize the result in a theorem, to make it clearer and more accessible, while leaving the annoying details on counting the single operations by hand to the supplementary material.
>
> > I am not sure about the comparison between BP and iPC in the experiments. Is the network architecture exactly the same? Normally with BP we have a feedforward architecture, without a generative interpretation.
>
> Yes, the architecture is always exactly the same in all the tasks that we have compared iPC against BP. When using generative models, we only compare PC against iPC, as shown in Fig. 2 (a,b).
>
>
> > Sometimes the term "Z-IL" appears but I think it was never defined. It appears more in the discussion. What is this?
>
> Addressed; thanks. Z-Il is a variation of PC that is able to exactly replicate the weight update of BP. We will clarify this in the revised version.
>
> > thinking about predictive coding as a model formulation (as I do, which I think is standard) vs. thinking about predictive coding as an optimization technique (as the paper does).
>
> This is a good point. We also think about predictive coding as a model formulation. However, iPC is indeed an optimization technique (i.e., a learning algorithm), specific for predictive coding networks. The efficiency results, for example, are obtained by training predictive coding networks using iPC. To make this concept clearer, we will refer to the models as “Predictive coding networks (PCNs)”, and to the update rule as PC or iPC. We have had a second read of the paper, and tried to make the difference among the two as clear as possible.

---

> > ### Author Response · Authors · 2022-11-11
> > **Table of runtimes 1/2**
> >
> > Here is the table of the official running times in milliseconds of all the experiments provided in Figure 3 of the paper:
> >
> > |   $L$ |   Hidden Dim. |   Seed |   iPC (ms) |      BP (ms) |
> > |------:|--------------:|-------:|-----------:|-------------:|
> > |     2 |             8 |   5341 |   19.8402  |     0.341654 |
> > |     2 |             8 |   7472 |    2.98715 |     0.141382 |
> > |     2 |             8 |   9273 |    3.9506  |     0.220537 |
> > |     2 |            16 |   5341 |    3.48687 |     0.144958 |
> > |     2 |            16 |   7472 |    3.92389 |     0.206947 |
> > |     2 |            16 |   9273 |    2.72107 |     0.147343 |
> > |     2 |            32 |   5341 |    4.7071  |     0.337601 |
> > |     2 |            32 |   7472 |    4.64439 |     0.257015 |
> > |     2 |            32 |   9273 |    4.23813 |     0.293732 |
> > |     2 |            64 |   5341 |    5.14531 |     0.521898 |
> > |     2 |            64 |   7472 |    3.50261 |     0.433445 |
> > |     2 |            64 |   9273 |    4.74596 |     0.578642 |
> > |     2 |           128 |   5341 |   10.8678  |     4.48537  |
> > |     2 |           128 |   7472 |   11.2545  |     4.80199  |
> > |     2 |           128 |   9273 |   10.3552  |     3.94964  |
> > |     2 |           256 |   5341 |   16.7456  |    16.1965   |
> > |     2 |           256 |   7472 |   17.9191  |    16.8622   |
> > |     2 |           256 |   9273 |   18.4166  |    16.2866   |
> > |     2 |           512 |   5341 |  160.162   |   251.189    |
> > |     2 |           512 |   7472 |  150.585   |   262.268    |
> > |     2 |           512 |   9273 |  160.136   |   262.373    |
> > |     2 |          1024 |   5341 |  595.119   |   997.341    |
> > |     2 |          1024 |   7472 |  597.221   |  1021.49     |
> > |     2 |          1024 |   9273 |  632.622   |  1147.89     |
> > |     4 |             8 |   5341 |    7.59339 |     0.28944  |
> > |     4 |             8 |   7472 |    6.31738 |     0.362158 |
> > |     4 |             8 |   9273 |    4.81153 |     0.312805 |
> > |     4 |            16 |   5341 |    3.72839 |     0.343561 |
> > |     4 |            16 |   7472 |    4.91095 |     0.319958 |
> > |     4 |            16 |   9273 |    3.79443 |     0.286818 |
> > |     4 |            32 |   5341 |    5.84936 |     0.500441 |
> > |     4 |            32 |   7472 |    6.40368 |     0.5126   |
> > |     4 |            32 |   9273 |    6.74725 |     0.51403  |
> > |     4 |            64 |   5341 |    5.96905 |     0.788212 |
> > |     4 |            64 |   7472 |    8.39949 |     0.929832 |
> > |     4 |            64 |   9273 |    7.04503 |     0.873327 |
> > |     4 |           128 |   5341 |   14.2429  |     8.55374  |
> > |     4 |           128 |   7472 |   16.2199  |     8.93378  |
> > |     4 |           128 |   9273 |   15.1129  |     8.7893   |
> > |     4 |           256 |   5341 |   23.9842  |    33.7701   |
> > |     4 |           256 |   7472 |   20.5846  |    33.2019   |
> > |     4 |           256 |   9273 |   36.7255  |    35.2798   |
> > |     4 |           512 |   5341 |  158.126   |   517.919    |
> > |     4 |           512 |   7472 |  167.869   |   497        |
> > |     4 |           512 |   9273 |  164.187   |   522.093    |
> > |     4 |          1024 |   5341 |  678.133   |  2015.67     |
> > |     4 |          1024 |   7472 |  664.752   |  2066.1      |
> > |     4 |          1024 |   9273 |  622.046   |  1997.78     |
> > |     8 |             8 |   5341 |    9.25779 |     0.61059  |
> > |     8 |             8 |   7472 |   10.5295  |     1.0767   |
> > |     8 |             8 |   9273 |    8.0502  |     0.564337 |
> > |     8 |            16 |   5341 |    7.69091 |     0.540972 |
> > |     8 |            16 |   7472 |   13.0358  |     0.863791 |
> > |     8 |            16 |   9273 |    8.44145 |     0.617743 |
> > |     8 |            32 |   5341 |    9.35507 |     1.15371  |
> > |     8 |            32 |   7472 |   10.0608  |     0.883102 |
> > |     8 |            32 |   9273 |   10.4787  |     1.03331  |
> > |     8 |            64 |   5341 |   10.4144  |     1.69182  |
> > |     8 |            64 |   7472 |   10.9556  |     1.62578  |
> > |     8 |            64 |   9273 |    7.75647 |     1.48463  |
> > |     8 |           128 |   5341 |   19.7504  |    15.8749   |
> > |     8 |           128 |   7472 |   15.4333  |    17.0178   |
> > |     8 |           128 |   9273 |   24.1373  |    19.2568   |
> > |     8 |           256 |   5341 |   35.5456  |    63.5219   |
> > |     8 |           256 |   7472 |   31.1813  |    63.3771   |
> > |     8 |           256 |   9273 |   32.109   |    62.4328   |
> > |     8 |           512 |   5341 |  231.559   |  1025.62     |
> > |     8 |           512 |   7472 |  266.099   |   982.408    |
> > |     8 |           512 |   9273 |  240.943   |  1085.49     |
> > |     8 |          1024 |   5341 |  870.819   |  4124.99     |
> > |     8 |          1024 |   7472 |  825.8     |  3995.9      |
> > |     8 |          1024 |   9273 |  804.621   |  4100.12     |

---

> > > ### Author Response · Authors · 2022-11-11
> > > **Table 2/2**
> > >
> > >
> > > |   $L$ |   Hidden Dim. |   Seed |   iPC (ms) |      BP (ms) |
> > > |------:|--------------:|-------:|-----------:|-------------:|
> > > |    16 |             8 |   5341 |   16.9411  |     1.18208  |
> > > |    16 |             8 |   7472 |   19.0938  |     1.08862  |
> > > |    16 |             8 |   9273 |   10.7536  |     1.01233  |
> > > |    16 |            16 |   5341 |   14.2412  |     0.998735 |
> > > |    16 |            16 |   7472 |   13.3479  |     1.64127  |
> > > |    16 |            16 |   9273 |   13.8955  |     1.00088  |
> > > |    16 |            32 |   5341 |   13.4552  |     1.90115  |
> > > |    16 |            32 |   7472 |   14.5161  |     1.6222   |
> > > |    16 |            32 |   9273 |   12.8353  |     1.41335  |
> > > |    16 |            64 |   5341 |   18.4188  |     3.74508  |
> > > |    16 |            64 |   7472 |   14.9627  |     2.6381   |
> > > |    16 |            64 |   9273 |   18.5626  |     3.79133  |
> > > |    16 |           128 |   5341 |   35.1024  |    30.864    |
> > > |    16 |           128 |   7472 |   35.4571  |    33.3576   |
> > > |    16 |           128 |   9273 |   31.0454  |    30.7331   |
> > > |    16 |           256 |   5341 |   55.4535  |   131.252    |
> > > |    16 |           256 |   7472 |   46.2825  |   134.253    |
> > > |    16 |           256 |   9273 |   46.9806  |   155.415    |
> > > |    16 |           512 |   5341 |  307.729   |  2071.97     |
> > > |    16 |           512 |   7472 |  355.52    |  2037.23     |
> > > |    16 |           512 |   9273 |  319.8     |  2073.53     |
> > > |    16 |          1024 |   5341 | 1327.06    |  9836.65     |
> > > |    16 |          1024 |   7472 | 1486.89    | 10146.4      |
> > > |    16 |          1024 |   9273 | 1421.91    | 10326.6      |
> > > |    32 |             8 |   5341 |   17.4241  |     3.14283  |
> > > |    32 |             8 |   7472 |   34.1778  |     3.09277  |
> > > |    32 |             8 |   9273 |   18.239   |     1.89447  |
> > > |    32 |            16 |   5341 |   22.4276  |     2.50888  |
> > > |    32 |            16 |   7472 |   18.6222  |     2.55203  |
> > > |    32 |            16 |   9273 |   25.3308  |     1.94812  |
> > > |    32 |            32 |   5341 |   24.7233  |     3.09539  |
> > > |    32 |            32 |   7472 |   20.9057  |     3.2227   |
> > > |    32 |            32 |   9273 |   21.1201  |     2.91514  |
> > > |    32 |            64 |   5341 |   20.0055  |     6.16384  |
> > > |    32 |            64 |   7472 |   25.9078  |     7.68113  |
> > > |    32 |            64 |   9273 |   21.2889  |     5.53656  |
> > > |    32 |           128 |   5341 |   60.7922  |    67.327    |
> > > |    32 |           128 |   7472 |   60.6296  |    66.6113   |
> > > |    32 |           128 |   9273 |   68.9101  |    91.809    |
> > > |    32 |           256 |   5341 |   77.8816  |   263.214    |
> > > |    32 |           256 |   7472 |   74.8813  |   289.155    |
> > > |    32 |           256 |   9273 |  110.585   |   252.253    |
> > > |    32 |           512 |   5341 |  558.599   |  4911.16     |
> > > |    32 |           512 |   7472 |  535.059   |  4991.77     |
> > > |    32 |           512 |   9273 |  628.008   |  4483.46     |
> > > |    32 |          1024 |   5341 | 2076.82    | 20333.9      |
> > > |    32 |          1024 |   7472 | 2112.99    | 20711.6      |
> > > |    32 |          1024 |   9273 | 2147.04    | 20581        |

---

> > ### Comment · Reviewer_fv42 · 2022-11-22
> > **Thank you**
> >
> > Thank you for the response. I think the paper is much stronger with these updates. Maybe the table of runtimes can be turned into a plot instead, but anyway the updates to the text are definitely improvements.
> >
> > I think the reviews are converging to the interpretation that: detached from the neuroscience influence, this is just incremental EM. Given that the neuroscience part is not really a bonus in the ICLR criteria, maybe the paper would be received better at another venue. I hope the authors are not discouraged by the lukewarm response overall, because (for me at least) the story and the "mix of two existing techniques" is still interesting.

---

### Official Review · Reviewer_kkJ5 · 2022-10-27

**Confidence:** 2
**Clarity, Quality, Novelty And Reproducibility:** This work is of good quality.
**Correctness:** 3
**Technical Novelty And Significance:** 3
**Empirical Novelty And Significance:** 3
**Recommendation:** 6

**Strength And Weaknesses:**

I do like the idea and the demonstrated performance of the proposed method. That being said, there are a few weaknesses I wish that the author could address:
* 3.2 on CPU implementation, I do not fully understand what is the overhead for iPC. Besides, the time complexity is related to L, then why does the improvement over BP also scales with hidden dimension (is this the amount of neurons in hidden layers)?
* For the experimental results reported in table 1, is the PC trained with the same amount of epochs as iPC, so that it behaves worse on AlexNet as it is not yet converges? The comparison with PC states that the iPC converges faster, but my understanding is that he iPC is a more efficient way to approximate PC, I do not understand why it failed to scale with model complexity. Besides, is it possible to test on some more complex model or image data set to back the idea that iPC preforms better on complex image related tasks?



**Summary Of The Paper:**

This paper proposes an incremental predictive coding method that performs faster on larger models than BP.

**Summary Of The Review:**

Overall, this is an interesting work. Some clarification on test results would make it more convincing.

---

> ### Author Response · Authors · 2022-11-11
> **Answer to reviewer**
>
> We thank the reviewer for his time.
>
> > why does the improvement over BP also scales with hidden dimension (is this the amount of neurons in hidden layers)?
>
>
> This is because of a computational overhead introduced by our implementation of iPC on CPU, and is hence merely an implementation drawback that should not be present in an ideal implementation of the algorithm. In that case, the improvement should scale only with the number of layers. We will make this more clear in the revised version.
> In our implementation, the overhead is given by the time needed to create multiple threads, where each thread runs the computation of one layer, and also the time of communication with these threads. The improvement over BP then scales with the hidden dimension (again, only for the considered implementation), as the overhead introduced by iPC is relatively fixed, and hence has a large impact when the hidden dimension is small, and a smaller one when the hidden dimension is large, as most of the time needed to perform the update is taken by the multiplication of two large tensors.
> For example, let us assume that the time needed to start L threads is 1 second, and the time needed to compute one multiplication in a small network is 0.01 seconds, iPC / BP is then (0.01+1)/(0.01*L)>1, so iPC is worse than BP, because of this fixed starting thread overhead. However, if we consider much larger layers, where the time needed to compute one operation is 1 second, the overhead becomes negligible, and we have that iPC / BP < 1.
>
>
>
> > is the PC trained with the same amount of epochs as iPC, so that it behaves worse on AlexNet as it is not yet converged?
>
> All the experiments in the table are performed using the same number of epochs. However, we have carefully checked whether the energy/loss of every model had converged, and this was indeed the case. Hence, the worse performance of PC on Alexnet is probably due to scaling properties of PC, rather than a non-converged network. This is a problem that we have not experienced using iPC, able to well scale to larger architectures.
> To conclude, note that iPC does not approximate PC. The convergence result states that both iPC and PC provably converge to a stationary point of the variational free energy, but it does not provide any approximation result as to the dynamics during training. Our extensive experiments, however, show that iPC tends to converge to better minima than PC. This is a solid result, as it has been tested on both generative and discriminative models on hundreds of different parametrizations (reported in both the supplementary material and main text). This is also reflected in the final test accuracies, that show that iPC tends to perform better than PC.

---

### Official Review · Reviewer_dzfS · 2022-10-27

**Confidence:** 3
**Correctness:** 3
**Technical Novelty And Significance:** 2
**Empirical Novelty And Significance:** 2
**Recommendation:** 5

**Clarity, Quality, Novelty And Reproducibility:**

The clarity, quality and reproducibility are mainly good (I spotted a few typos - for instance, in Eq. 4, the conditional in the second expression should read 'p(x^(l-1) | x^(l))', and the Gaussian formulation in the third expression should include the prior and the x's).  As noted above, the novelty is an issue for me.

**Details Of Ethics Concerns:**

None.

**Strength And Weaknesses:**

Strengths:

-  The biological plausibility argument is interesting, and in general the argument is convincing that some form of 'localized EM' algorithm is more plausible than BP or PC alternatives, while retaining convergence and generalization properties.
-  The experimentation convincingly demonstrates that iPC should be considered a viable alternative to BP generally, at least for simple architectures and specialized hardware.

Weaknesses:

- I'm mainly concerned about the paper's novelty - essentially, iPC is equivalent to iEM applied to a hierarchical Gaussian model.  The theoretical properties are described elsewhere (e.g. Karimi 2019) and the biological plausibility argument is hard to evaluate, although likely to be worth pursuing further.
- There is little theoretical novelty, since the time-complexity analysis (Theorem 1) essentially follows simply by definition.  As discussed by the authors, the comparison of training-loss convergence rates in terms of 'non-parallel matrix multiplications' is an interesting result, but this is investigated solely empirically (Fig. 2 right).

**Summary Of The Paper:**

This paper describes a variant of predictive coding, named incremental predictive coding (iPC), based on incremental EM, which it is argued should be considered a biologically plausible approach to learning in the brain.  The complexity of iPC is considered in relation to back-propagation (BP), and a CPU implementation is provided.  Further, the generalization performance is investigated on a number of datasets, and the algorithm is shown to perform well in comparison to BP and PC.

**Summary Of The Review:**

An interesting investigation of an algorithm that may have relevance in neuroscience, and deserves further attention.  Potentially, the paper may be of interest to those working in neuroscience and optimization.

---

> ### Author Response · Authors · 2022-11-11
> **Answer to the reviewer**
>
> We thank the author for the feedback on our work.
>
> Regarding the weaknesses of the paper:
>
> > I'm mainly concerned about the paper's novelty - essentially, iPC is equivalent to iEM applied to a hierarchical Gaussian model.
>
> We agree that iPC is obtained by applying incremental EM to the original formulation of PC, and it is hence a mix of two existing techniques. This, however, does not affect the novelty of the proposed algorithm, as it addresses a fundamental problem in the field. The first work introducing the weight update of PC is Rao and Ballard’s, in 1999. The incremental EM is from Neil & Hinton, in 1996. From Rao and Ballard’s paper to today, countless works use the original rule to update the weights, that is much slower than ours, and less performing in the tasks we have proposed. The authors performing the experiments have certainly experienced the bad efficiency of PC models. Hence, as obvious as the proposed algorithm can be, it certainly addresses an existing, practical, problem that a large community has had for years. The fact that it empirically converges to a better minimum (shown in a large number of experiments in this paper) is also a big plus.
>
> As your main concern is the novelty of this work, we point to the ‘On Novelty” paragraph that we have added as an answer to all the reviewers.
>
>
> > The comparison of training-loss convergence rates in terms of 'non-parallel matrix multiplications' is an interesting result, but this is investigated solely empirically
>
> Note that in a classification task, it is possible to initialize all the internal variables of the model to have zero error. In this case, at t=0, the total energy of the model would be concentrated in the output layer, and hence be equivalent to the train loss. This has been shown to lead to a good performance empirically. Starting from this assumption, we can use the iEM algorithm to minimize the total energy of the network, equivalent in this case to the  the train loss.
>
> > Typos
> Thank you for the pointers, we have addressed them.

---

> > ### Comment · Reviewer_dzfS · 2022-12-13
> > **Response**
> >
> > Thanks to the authors for their response.  I agree with many of the points in the response, particularly that this approach is a more biologically plausible algorithm for PC as opposed to Rao and Ballard, and that the simplicity of the approach here is a positive feature (potentially enhancing its biological plausibility).  I agree that this observation can be considered novel, particularly from a neuroscience perspective, and I'd be interested to see further work along these lines.

---

### Decision · Program_Chairs · 2023-01-20

**Decision:**

Reject

**Justification For Why Not Higher Score:**

See reviews. This paper may contain novelties in some neuroscience circles, but not enough for ICLR.

**Justification For Why Not Lower Score:**

N/A

**Metareview: Summary, Strengths And Weaknesses:**

This paper points out that a learning algorithm called "predictive coding" in neuroscience is essentially maximum likelihood estimation in a basic latent variable model. As is well known, ML estimation in such models can conveniently be done with the EM algorithm, even though sometimes direct gradient-based learning can be faster. The authors then suggest to apply Neal and Hinton's classical incremental EM, and point out some advantages of that, in particular scalability to large data.

Neal and Hinton's incremental EM is well known in the probabilistic ML field, where ICLR is situated, even though this may not be the case for certain neuroscience circles, where maybe the proposed work would be better placed. For ICLR readers, there is simply nothing much new here. Looking at the call for papers, while ICLR encourages submission of "neuroscience applications", debates about "biological plausibility" and such are probably better placed elsewhere.

I'd also like to state that the interaction with the authors has been somewhat painful, and would strongly recommend to them to change their tone when interacting with AC and reviewers who may have different opinions to them. Use of capital letter words and strong claims are not appreciated, and probably less than helpful as a response to feedback.